

# Synthesis of nano-TiO$_2$ assisted by glycols and submitted to hydrothermal or conventional heat treatment with promising photocatalytic activity

Lidiaine Maria Santos[1] and Antonio Eduardo da Hora Machado[2,3]

[1] Department of Chemistry, Instituto Federal de Educação, Ciência e Tecnologia de Goiás (IFG), Uruaçu, Goiás, Brazil
[2] Laboratory of Photochemistry and Materials Science –LAFOT-CM Instituto de Química, Universidade Federal de Uberlândia, Uberlândia, Minas Gerais, Brazil
[3] Graduate Program in Exact and Technological Sciences, Universidade Federal de Catalão, Catalão, Goiás, Brazil

## ABSTRACT

TiO$_2$ nanoparticles were successfully synthesized by the sol-gel method employing different glycols (ethylene glycol, diethylene glycol or polyethylene glycol 300), which were heat-treated in conventional oven or by hydrothermal via, obtaining photocatalysts with particle sizes and distinct crystalline structures. HRTEM analyses showed that the oxides submitted to hydrothermal treatment featured spherical morphology, being formed by partially aggregated particles with sizes varying between 2 and 5 nm. X-ray diffractograms and Raman spectroscopy confirm that anatase was predominant in all synthesized compounds, with presence of brookite phase for samples that received hydrothermal treatment or were synthesized in the presence of polyethylene glycol with heat treatment in conventional oven. The amount of brookite as well as the cell volume, deformation, network parameters and crystallinity were estimated by Rietveld refinement. The surface area and porosity of the materials were higher when the synthesis involved the use of hydrothermal treatment. These oxides are mesoporous with porosity between 14 and 31%. The oxide synthesized in the presence of ethylene glycol with hydrothermal thermal treatment (TiO$_2$G1HT) exhibited the highest photocatalytic activity in terms of mineralization of azo-dye Ponceau 4R (C.I. 16255), under UV-Vis irradiation. This higher photocatalytic activity can be attributed to the formation of binary oxides composed by anatase and brookite and by its optimized morphological and electronic properties.

## INTRODUCTION

Titanium dioxide (TiO$_2$) is widely employed in technological applications, including solar energy conversion, chemical sensors for gases, environmental depollution and hydrogen production, among others (*Machado et al., 2015*; *Riyapan et al., 2016*). It is an n-type semiconductor, with band gap energy of the extended solid (bulk) in the ultraviolet region

Corresponding author
Lidiaine Maria Santos,
lidiaine.santos@ifg.edu.br

of approximately 3.20, 3.02 and 3.14 eV, respectively for anatase, rutile and brookite, the three natural polymorphs (*Grätzel & Rotzinger, 1985*).

TiO$_2$-based materials are the most investigated for photocatalytic application since the discovery by Fujishima e Honda (*Fujishima & Honda, 1972*). It is one of the most commonly used semiconductor oxide for environmental photocatalysis, being of low toxicity, insoluble in water and stable to photo and chemical corrosion over a wide range of pH (*Machado et al., 2012*; *Rekulapally et al., 2019*). The photocatalytic process involves the electronic excitation from the valence (VB) to the conduction band (CB), when irradiated by ultraviolet light. This process generates charge carriers ($e^-/h^+$ pairs) that react with molecular oxygen and water, forming reactive oxygen species (ROS), such as superoxide radical ion ($O_2^{\cdot-}$) and hydroxyl radical (HO·). These and other secondary-generated radicals promote the degradation of environmental pollutants (*Machado et al., 2008*; *Muthamizhchelvan et al., 2020*).

Anatase, the most active polymorph for photocatalytic applications, contains more defects due to its structure and acts as electron trap (*Gupta & Tripathi, 2011*). Already in rutile, there is a high $e^-/h^+$ recombination rate, which limits the photocatalytic response. On the other hand, the brookite photocatalytic activity seems to be related to the relative position of the electronic bands, where CB energy is 0.14 eV more negative than that of anatase in anatase/brookite associations, favoring the photocatalytic processes (*Kandiel et al., 2010*; *Patrocinio et al., 2015*).

The physical and chemical properties of TiO$_2$ depend on the arrangement of the crystalline phase, size and shape of the particles, surface area and crystallinity (*Tan, Sato & Ohara, 2015*; *El-Sheikh et al., 2017*), parameters that can be controlled or adjusted during the synthesis process.

This oxide can be obtained by different synthetic routes (*Ahmadi et al., 2015*; *Benetti et al., 2016*; *Hajizadeh-Oghaz, 2019*; *Kumar et al., 2019*). The synthesis via sol–gel methodology can be improved by the use of reagents with long hydrophobic chains favoring the controlled formation of critical nuclei, leading to the obtaining of mesoporous particles in a nanometric scale (*Wang et al., 2014*; *Darbandi & Dickerson, 2016*; *El-Sheikh et al., 2017*; *Catauro et al., 2018*)). Crystallization is a necessary step for obtaining oxides with a defined structure, purity and desirable morphology. This process can occur either by conventional heat (*He et al., 2014*; *Patrocinio et al., 2015*) or by hydrothermal treatment (*Kim & Kwak, 2007*; *Qin et al., 2016*).

The application of TiO$_2$ as a photocatalyst has some disadvantages that can be overcome. The main disadvantage is the high band gap, followed by the relatively high recombination rate of the charge carriers, this last reducing considerably the quantum yield of the photocatalytic processes (*Kumar & Devi, 2011*; *Feng, Zhang & Yu, 2012*; *Jaiswal et al., 2015*; *França et al., 2016*).

The existence of junction between different phases of the same semiconductor, as for example in anatase/brookite (A/B) or anatase/rutile (A/R) mixtures (*Cihlar et al., 2015*), results in synergistic effects that leads to a more efficient separation of the $e^-/h^+$ pairs, reducing the charge recombination rate. Consequently, while the electrons are trapped in one of the crystalline phases, the holes present in the VB have greater chance to oxidize

organic matter, enhancing the photocatalytic activity (*Yang et al., 2013*; *Shao et al., 2014*; *Zhang et al., 2020*).

In the present study, photocatalysts based on $TiO_2$ were synthesized by the sol–gel method. The influence of the use of different structural molds (ethylene glycol, diethylene glycol or polyethylene glycol 300) as well as the effect of thermal treatments by conventional or hydrothermal routes, was evaluated on their photocatalytic activity, and structural optical and morphological properties. The photocatalytic activity was evaluated through the degradation of the azo-dye Ponceau 4R, chosen due to its industrial application and undesirable effects on the environment and human health (*Oliveira et al., 2012*; *EuropeanFoodSafetyAuthority, 2020*). The results presented here aim to provide new insights into the synthesis of $TiO_2$-based photocatalysts with different crystalline phases and the influence of preparation conditions on the photocatalytic properties of these systems.

## MATERIALS & METHODS

### Preparation of different $TiO_2$ photocatalysts

All chemicals were of analytical or HPLC grade and were used as received. Ultrapure water obtained from an Elix 5 Milli-Q® water purification system was employed in all experiments. $TiO_2$ samples were synthesized by the sol–gel method, using different glycols (ethylene glycol, diethylene glycol or polyethylene glycol 300) (Sigma Aldrich), and heat treatment in a conventional oven or hydrothermal system.

The $TiO_2Gx$ photocatalyst was obtained from the mixture, under magnetic stirring, of 10 mL of Ti (IV) isopropoxide (Aldrich, 97%) and 50 mL of glycol (where $x = 1$ when 886 mmol of ethylene glycol (Vetec, 99.5%) were used, $x = 2$ for 527 mmol diethylene glycol (Vetec, 99.5%), and $x = 3$ when polyethylene glycol 300 (Fluka) was used). After 2 h of stirring, a mixture containing 10 mL of ultrapure water and 90 ml of acetone (Synth, 99.5%) was added to the suspension and kept under stirring for 2 h. The white precipitate was separated with the aid of a centrifuge (9,000 rpm for 20 min), followed by washing several times with ethanol to remove residues of glycol, followed by washing three times with distilled water.

For the preparation of heat-treated photocatalysts in a conventional oven ($TiO_2GxM$), after washing the powder was dried at 70 °C under reduced pressure and sintered at 400 °C for 2 h. After centrifugation and washing the decanted oxide prepared using hydrothermal treatment, $TiO_2GxHT$, was submitted to the hydrothermal reactor under a pressure of approximately 13.8 bar at 200 °C for 4 h. Subsequently, it was dried at 70 °C for 24 h.

### Characterization of the photocatalysts

High resolution electronic transmission images were obtained using a Jeol, JEM-2100, Thermo scientific Transmission Electron Microscope. The particle size and spacing between crystalline planes were calculate with the free software "ImageJ".

X-ray diffraction analyses (XRD) using a Shimadzu XRD600 powder diffractometer operating at 40 kV and 120 mA, employing Cu K $\alpha$ ($\lambda$= 1,54148 Å) radiation. The diffractograms were collected between $10° \leq 2\theta \leq 90°$ under a rate of 0.5° $min^{-1}$.

Crystalline silicon was used as the diffraction standard. X-ray diffratogram of the oxides were refined by the method of Rietveld using the FullProf software, with fitting criteria (Factor S - Goodness of Fit) was employed as the ratio between the weight factor ($R_{wp}$) and the expected factor ($R_{exp}$), which should be closer to 1. The fit parameters can be found in the (Table S1).

N$_2$ adsorption–desorption isotherms were obtained using an ASAP 2010 analyzer (Micrometrics). The specific area were analyzed using the Brunauer, Emmett and Teller (BET) model and the Barrett-Joyner-Halenda (BJH) model for the porous volume (*Barrett, Joyner & Halenda, 1951*).

Raman spectra were acquired at room temperature using a Bruker spectrometer model RFS 100/S, samples were excited at 1064 nm with laser operating at 100 mW. Diffuse reflectance spectra of the synthesized oxides were acquired using an UV-1650PC Spectrometer (Shimadzu), at room temperature and potassium bromide was used as reference. The band gap energy being estimated by the Kubelka–Munk function (*Patterson, Shelden & Stockton, 1977*).

### Photocatalytic activity

In all photocatalytic assays, 100 mg L$^{-1}$ of the catalyst was added to 31 mg L$^{-1}$ dye Ponceau 4R (trisodium (8Z)-7-oxo-8-[(4-sulfonatonaphthalen-1-yl)hydrazinylidene]naphthalene-1,3-disulfonate, CI 16255, Sigma-Aldrich, 75%) aqueous solution (pH = 6.9) under magnetic stirring. The experimental setup was previously described in detail (*Oliveira et al., 2012*). Information about the radiation source and experimental data were available in (*Machado et al., 2008*; *Santos et al., 2015*).

The photocatalytic system was kept at 40 ± 2 °C and under stirring for 30 min in the dark to reach the adsorption equilibrium. Control measurements in the dark were performed and in the absence of a catalyst to evidence the role of TiO$_2$ in the photochemical reaction. Aliquots were taken at 20 min intervals, filtered and analyzed by spectrophotometry, following the discoloration at 507 nm using a Shimadzu spectrophotometer model 1650PC and by Total Organic Carbon (TOC) measurements, using a Shimadzu TOC-VCPH/CPN analyzer.

## RESULTS & DISCUSSION

TEM images evidence that the heat-treated oxides in a conventional oven and by hydrothermal via are made up of approximately spherical nanoparticles (Figs. 1A 1D). For the material produced after processing in a conventional oven, the formation of agglomerates was more pronounced than after hydrothermal treatment. The estimated particle size from the HRTEM images (Figs. 1E–1H) were 10 nm, 2 nm, 3 nm e 4 nm respectively for the oxides TiO$_2$G1M, TiO$_2$G1HT, TiO$_2$G2HT e TiO$_2$G3HT. The sample calcined in a conventional oven, TiO$_2$G1M, presented the largest particle size, probably due the coalescence process by diffusion of smaller (more unstable) particles, favoring the formation of agglomerates. The formation of smaller, although stable, particles was observed after hydrothermal treatment, which tends to produce particles with larger surface

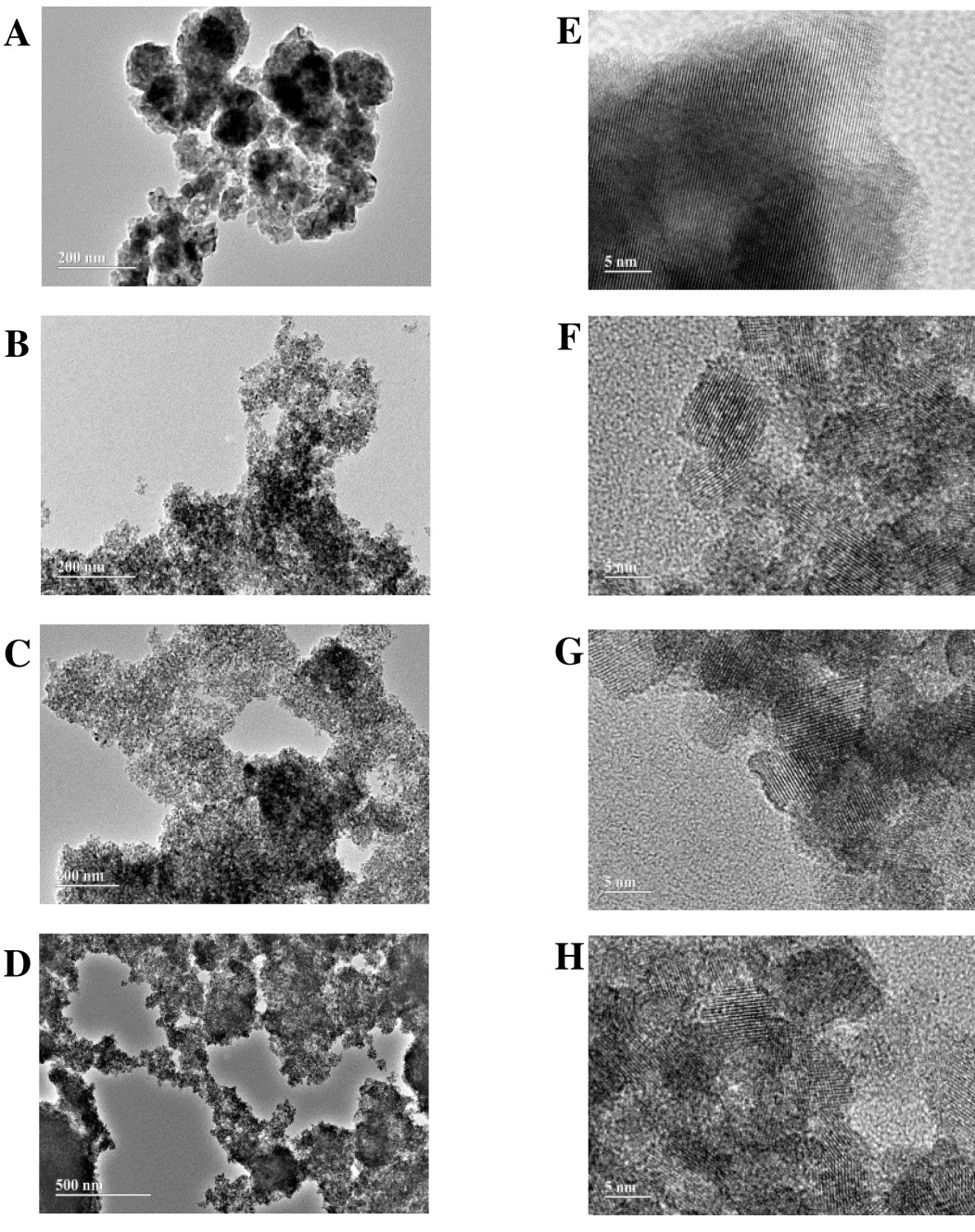

**Figure 1**  TEM and HRTEM images of mesoporous TiO$_2$: (A, E) TiO$_2$G1M, (B, F) TiO$_2$G1HT, (C, G) TiO$_2$G2HT, (D, H) TiO$_2$G3HT.

areas. The use of different glycols in the synthesis also influenced the particle size due the increase in the carbon chain (G1 <G2 <G3), which resulted in slightly larger particles.

HRTEM images suggest high crystallinity, mainly for the sample treated in a conventional oven. The spacing between crystalline planes, for all samples, was estimated as being 0.35 nm, corresponding to the (101) plan of the anatase phase, indicating that in all cases

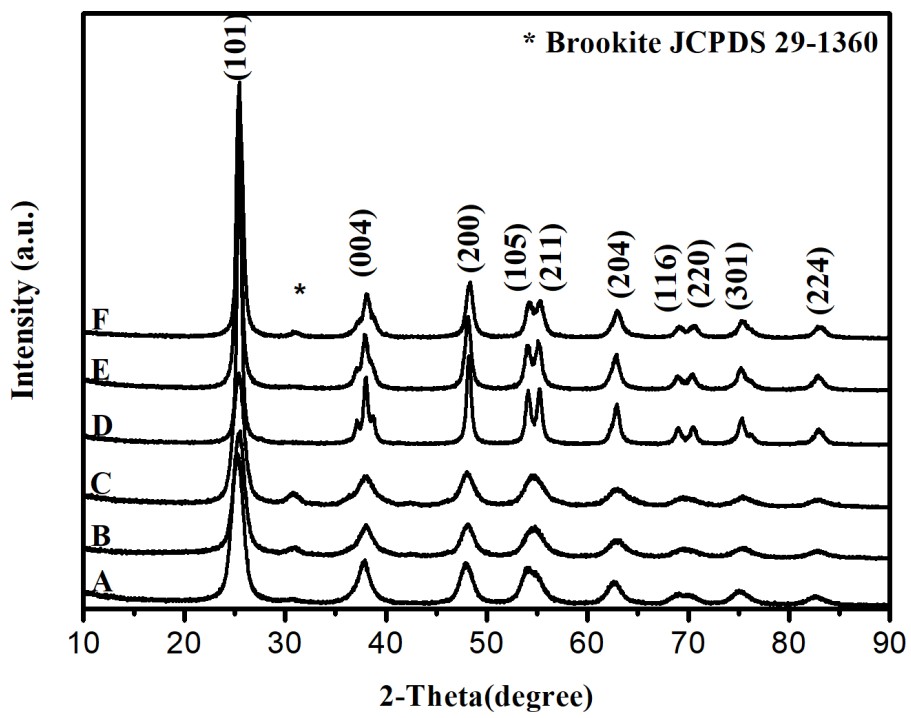

**Figure 2** X-ray diffraction patterns of the studied photocatalysts: (A) TiO$_2$G1HT; (B) TiO$_2$G2HT and (C) TiO$_2$G3HT, (D) TiO$_2$G1M; (E) TiO$_2$G2M and (F) TiO$_2$G3M.

this is the preferential growth plan for nanoparticles. The presence of crystallographic planes referring to the brookite and rutile crystalline phases were not verified through the images. In the case of brookite, the spacing between the crystallographic planes, of 0.38 nm, corresponding to the (120) and (111) planes, is very close to the spacing of the anatase (0.35 nm) causing ambiguity (*Kobayashi, Petrykin & Kakihana, 2007*). These plans, in the presence of higher anatase content, are overlaid by the (101) plan. The spacing between the 0.29 nm crystallographic planes, corresponding to the (121) plane, characteristic of the brookite, was not observed (*Di Paola, Bellardita & Palmisano, 2013*)

The XRD data (Fig. 2) confirm that all samples are composed mostly of nanocrystals of anatase, with the (101) phase preferably exposed. In the case of HT processing at 200 °C for 4 h, the presence of crystalline anatase phases and traces of brookite was observed, being confirmed by the presence of peaks at $2\theta$ equal to 25.38° (101) and 30.80° (121), respectively. Under the treatment conditions to which these materials were submitted, the formation of the rutile phase was not observed. The formation of the brookite phase was probably a crucial factor for the inhibition of the transformation of anatase into rutile.

Rietveld analyses of the diffractograms (Fig. S1) confirm the decrease in crystallite size for the oxides obtained after hydrothermal heat treatment (HT), which agrees with the HRTEM images. These quantitative data also confirm the greater presence of brookite phase in samples submitted to hydrothermal treatment. Besides that, it is observed that the percentage of the brookite phase remains practically constant even with the use of different

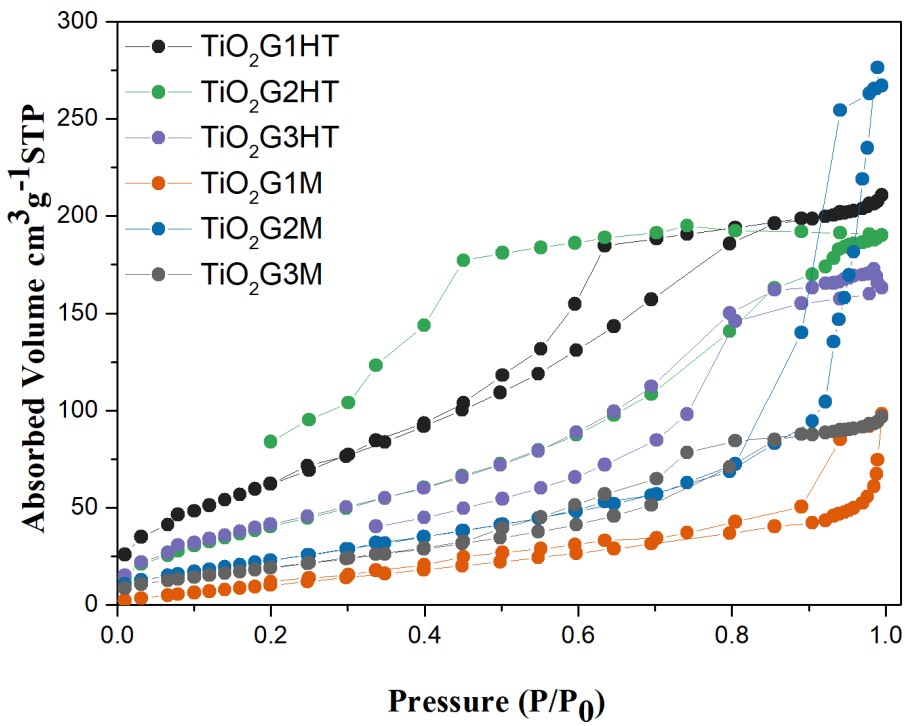

**Figure 3** N$_2$-adsorption–desorption isotherms obtained for the photocatalysts.

glycols in the synthesis process. Already for materials prepared with heat treatment in a conventional oven, it turns out that the use of different glycols leads to greater deformations only for the TiO$_2$G3M sample, where polyethylene glycol was used in the synthesis, causing the formation of 17.47% brookite phase (*Tay et al., 2013*).

The TiO$_2$G3HT sample had a higher portion of brookite when compared to the TiO$_2$G3M sample, because since brookite is featured by its low symmetry, its formation is more efficient under mild conditions such as shorter period and lower preparation temperature, as occurs in hydrothermal treatment conditions (*Lin et al., 2012*).

The formation of mesoporous structures was confirmed by N$_2$ adsorption–desorption isotherms (Fig. 3). Isotherms follow the type III for samples with 100% anatase phase (TiO$_2$G1M e TiO$_2$G2M). The other samples, with brookite content, have type IV with a pronounced hysteresis loop of types H3 and H4, according to the IUPAC classification. This suggests that these materials are mesoporous solids formed by agglomerated or aggregated particles (*Gregg & Sing, 1982*). The presence of brookite causes a decrease in the average pore diameters, suggesting that the presence of structural defects influences the adsorption capacity and porosity of the material. The values of surface area and porosity of these materials are presented in Table 1.

The diffuse reflectance spectra, expressed in terms of F(R) vs. photon energy (E), are presented in Fig. 4. The indirect band gap value ($E_g$) was obtained by extrapolating the linear segment to the X axis, Table 1. However, a simple inspection of the spectra suggests that the band gap values calculated in this way calculated in this way are deviated from

**Table 1  Morphologic and electronic parameters to oxides synthesized.**

| Photocatalysts | Surface area ($m^2g^{-1}$) | Porosity (%) | $E_g$ (eV) | $E_{g(real)}$ (eV) |
|---|---|---|---|---|
| TiO$_2$G1HT | $240.0 \pm 4.7$ | 30.6 | 3.30 | 2.64 |
| TiO$_2$G2HT | $158.5 \pm 2.9$ | 27.8 | 3.23 | 2.87 |
| TiO$_2$G3HT | $161.2 \pm 2.4$ | 25.1 | 3.28 | 3.05 |
| TiO$_2$G1M | $51.8 \pm 2.2$ | 12.4 | 3.27 | 2.86 |
| TiO$_2$G2M | $90.4 \pm 2.3$ | 12.3 | 3.25 | 3.00 |
| TiO$_2$G3M | $74.6 \pm 1.8$ | 14.1 | 3.24 | 3.01 |

the actual values, since the radiation absorption is not canceled ($E < E_g$), except from the point where F(R) $\rightarrow$ 0. This suggests the existence of permitted states with energies lower than the estimated $E_g$, that is, $E_{g(real)} < E_g$. Thus, considering the lower threshold of the conduction band, which occurs when F(R) $\rightarrow$0, that is, states with energies less than or equal to the energy associated with this threshold, are prohibited. In view of this, $E_{g(real)}$ was also calculated (Table 1). Based on this information, it appears that all photocatalysts absorb radiation more intensely in the near-UV region. However, these photocatalysts, despite the high band gap energies, have significant photocatalytic activity in the visible region, as suggest the estimated values of $E_{g(real)}$. The TiO$_2$G1HT, TiO$_2$G2HT and TiO$_2$G1M photocatalysts show a radiation absorption profile shifted to the visible region, with $E <E_g$, being therefore able to uptake photons in a large range of wavelengths. Related to these factors, the high surface area, crystallinity and mixture of crystalline phases are added, which end up favoring the photocatalytic potential of these oxides.

The electronic properties of the particles change significantly by reducing their size. Thus, new properties can be expected in nanoparticles when compared to bulk (*Hodes, 2007*. The variation of energy as a function of size promotes the quantum confinement and is characterized by an increase in the indirect band gap energy ($E_g$), as can be seen for TiO$_2$G1HT, which has smaller particle and crystallite sizes, as estimated by HRMET and DRX analyses, and $E_g$ (3.30 eV) greater than that of the extended solid (3.20 eV for TiO$_2$) (*Kumar & Devi, 2011*).

The catalysts were also evaluated using Raman spectroscopy (Fig. 5). All samples exhibit vibration modes typical of anatase ($3E_g + 2B_{1g} + A_{1g}$). $A_{1g}$ symmetry mode was not visualized, probably due the overlap with the band corresponding to the second mode, of $B_{1g}$ symmetry (*Iliev, Hadjiev & Litvinchuk, 2013; Fang et al., 2015*). A slight change in the signs is observed depending on the type of heat treatment used (Fig. 5 - Inset). The bands referring to samples thermally treated by hydrothermal route are broader than those observed for the calcined oxides in a conventional oven. This broadening can be directly correlated to the concentration of oxygen vacancies on the photocatalysts, as previously shown by (*Parker & Siegel (1990)*. Thus, Raman analysis indicates that the synthesis of oxides treated by the hydrothermal route, induces the formation of oxygen vacancies on the oxide surface, increasing the system disorder.

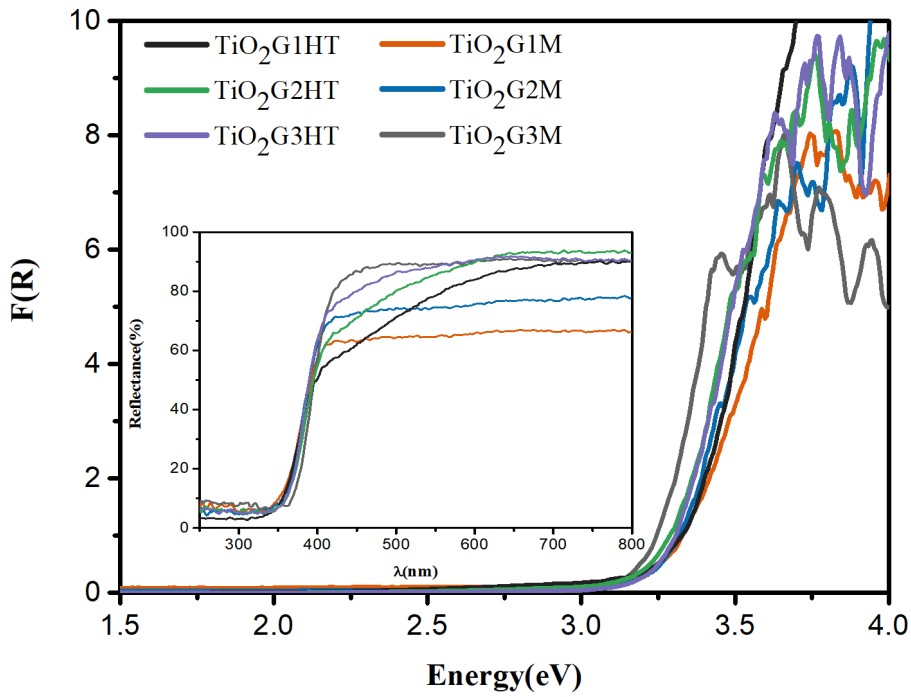

**Figure 4 Diffuse reflectance spectra of the $TiO_2$ photocatalysts.** Inset: Diffuse reflectance spectra of the samples.

## Photocatalytic activity

The photocatalytic activity of the different synthesized oxides was evaluated in terms of the degradation of the azo-dye Ponceau 4R. The control experiment, in the absence of any photocatalyst, reveals extremely low levels of dye discoloration (4.0%) and mineralization (13%) after 140 min of irradiation (Fig. S2). The degradation efficiency presented by the different photocatalysts is summarized in Table 2.

The oxides thermally treated by hydrothermal via were more efficient than conventional heat treatment in promoting the degradation and mineralization of the dye under study. Calcination in a conventional oven led to an increase in the crystallinity of the materials, as seen by the XRD data, and a decrease in the surface area, which ended up compromising the photocatalytic activity of these oxides.

The photocatalytic performance exhibited by the samples synthesized in the presence of different glycols and thermally treated by hydrothermal via can be attributed to the coexistence of anatase and brookite the high surface area, mesoporosity, and more appropriate particle sizes. Crystalline materials with smaller particle sizes are more likely to exhibit expressive photocatalytic properties (*Ohno et al., 2001*).

Although the $TiO_2$G3M photocatalyst also presents itself as a mixture of polymorphs anatase and brookite, it did not show significant photocatalytic activity, probably related to its smaller surface area.

The increase in photocatalytic activity of samples that present anatase and brookite can be explained by the synergism between these polymorphs. Although anatase and

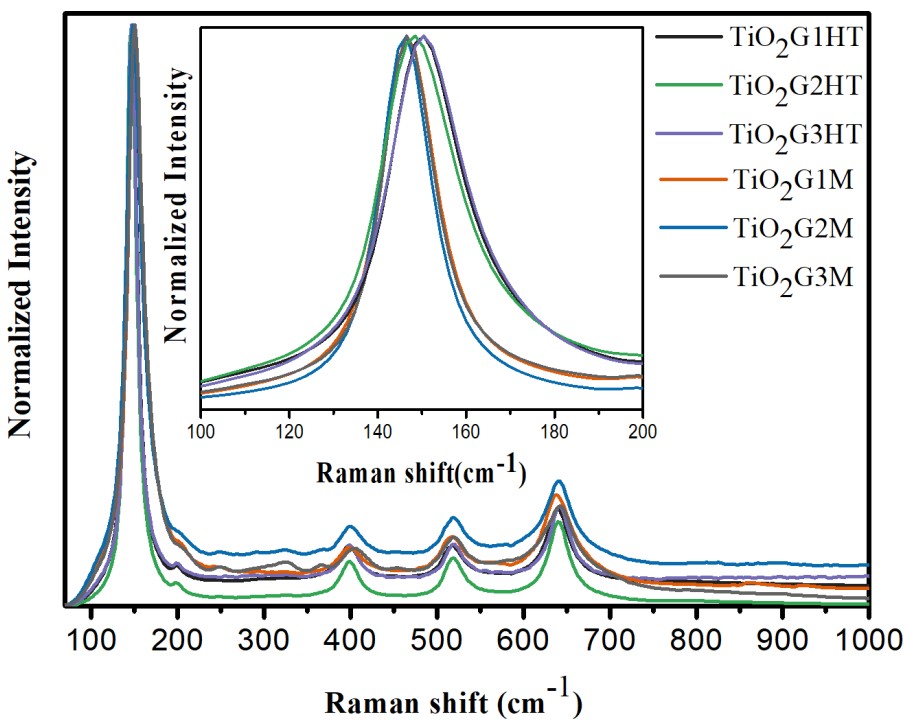

**Figure 5** **Raman spectra, at room temperature, for the synthesized TiO₂ photocatalysts.** Inset: expanded normalized Raman spectra between 100 and 200 cm$^{-1}$ in the main E$_g$ peak region attributed to the broadening of the band according to the type of heat treatment.

**Table 2** **Mineralization, LangmuirHinshelwood kinetics and discoloration of the P4R azo-dye mediated by the prepared oxides.**

|  | TiO₂G1HT | TiO₂G2HT | TiO₂G3HT | TiO₂G1M | TiO₂G2M | TiO₂G3M |
|---|---|---|---|---|---|---|
| Mineralization (%) | 58 | 44 | 46 | 23 | 24 | 17 |
| $k_{app}$/($\times$ 103 min$^{-1}$) | 5.9 | 4.3 | 4.4 | 1.7 | 1.9 | 1.5 |
| Discoloration (%) | 100 | 100 | 100 | 70 | 74 | 67 |

brookite present a very close $E_g$ (*Machado et al., 2012*; *Patrocinio et al., 2015*), theoretical calculations have shown that the energies of the conduction and valence bands of anatase phase are slightly lower than the corresponding energy levels of brookite (*Li et al., 2008*), suggesting a certain ease of migration of electrons from brookite to anatase. Thus, the holes are more available for oxidation reactions. In addition, the energy barrier between these polymorphs will tend to hinder the recombination among charge carriers. Therefore, with an extended life span, holes in the brookite valence band have a greater chance to oxidize organic matter, while electrons "trapped" in anatase may favor reduction reactions, leading to an increase in the photocatalytic activity (*Li, Ishigaki & Sun, 2007*; *Patrocinio et al., 2015*).

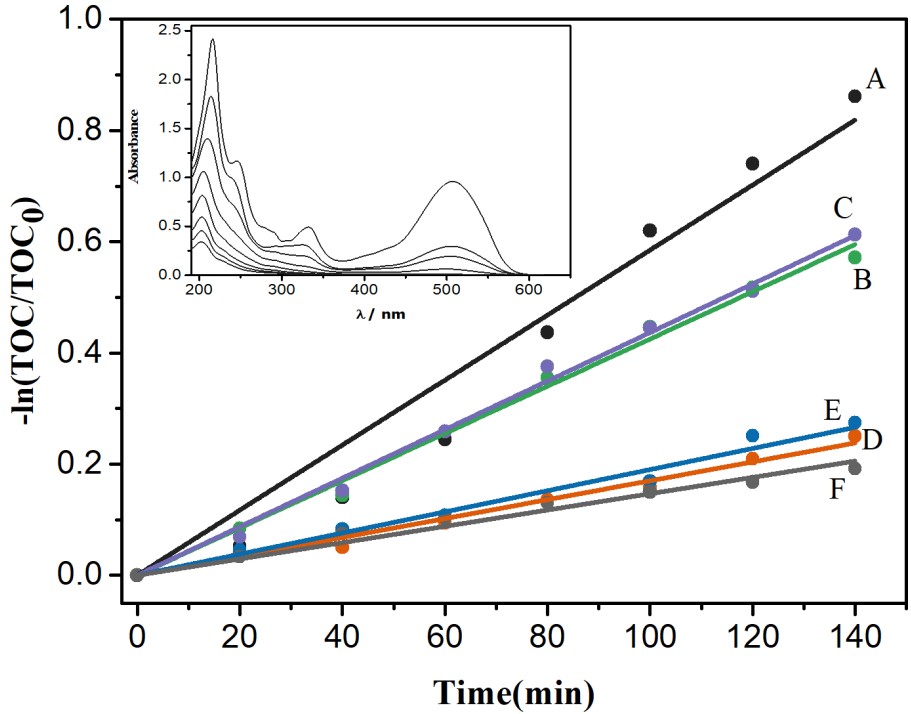

**Figure 6** **P4R mineralization kinetics using different photocatalysts: $TiO_2$ G1HT (A), $TiO_2$ G2HT (B), $TiO_2$ G3HT (C), $TiO_2$ G1M (D), $TiO_2$ G2M (E) and $TiO_2$ G3M (F).** Inset: absorption spectrum of the dye solution as a function of the irradiation time ($\Delta t = 20$ min), during the action of the $TiO_2$G1HT photo-catalyst.

A complex degradation mechanism is expected in heterogeneous photocatalysis (*Hoffmann et al., 1995*; *Ahmed et al., 2010*). The reactions occur initially at the solid-solution interface and involve reactive species generated on the surface of the excited photocatalyst or by direct interaction between the excited photocatalyst and the substrate (*Oliveira et al., 2012*; *Santos et al., 2015*). In the degradation under study, the discoloration of the dye is probably related to the homolytic scission of the azo group. Hydroxyl radicals (HO·), formed in the solid-solution interface, may be responsible for this process (*Kumar & Devi, 2011*). Table 2 presents data on the percentage of discoloration in the reactions mediated by the oxides synthesized in this study. The best performances occurred using oxides submitted to hydrothermal treatment.

The mineralization process follows a Langmuir–Hinshelwood kinetics (*Hoffmann et al., 1995*), being of pseudo-first order in relation to the dye, as show in Fig. 6. The rate constants are listed in Table 2.

In these assays, it was found that only 4.0% of the dye was adsorbed in the photocatalyst (Fig. S2), suggesting that the observed mechanism occurs mainly through the photodegradation of organic matter, certainly by the action of reactive oxygen species (ROS), such as HO· or $O_2^{·-}$, with the predominant action of the HO· radicals, a very strong oxidizing agent (standard reduction potential of $HO^·/H_2O$ 2.38 $V_{vs.}$ NHE) (*Hoare, 1985*). Accordingly, and based on the characterization of the photocatalysts, we can propose a

mechanism, shown in Eqs. (1)-(9) which must occur at the solid-solution interface, where $TiO_2$ (A) is the anatase polymorph and $TiO_2$ (B) is the brookite polymorph. As result of the photoexcitation of the catalyst (1), the $e^-/h^+$ pairs are generated; recombination processes (2) compete with the electron trapping in the polymorph anatase (3) and holes in the brookite polymorph (4 and 5), generating the reactive species responsible for the degradation of the dye (5, 6 and 7). In the valence and conduction bands, the oxidation (8) and reduction (9) reactions occur, respectively, resulting in degradation products.

$$TiO_2(A+B)+hv(UV) \rightarrow TiO_2(e_{cb}^- + h_{vb}^+) \tag{1}$$

$$e^-(cb) + h^+(vb) \rightarrow TiO_2(A+B) + hv(or\ heat) \tag{2}$$

$$TiO_2(A)(e_{cb}^-) + O_2 \rightarrow TiO_2 + O \tag{3}$$

$$TiO_2(B)(h_{vb}^+) + H_2O \rightarrow TiO_2 + H^+ + HO^\bullet \tag{4}$$

$$TiO_2(B)(h_{vb}^+) + OH^- \rightarrow TiO_2 + HO^\bullet \tag{5}$$

$$O_2^{-\bullet} + H^+ HO_2^\bullet \tag{6}$$

$$P4R + OH^\bullet \rightarrow Degradation\ products \tag{7}$$

$$P4R + h_{vb}^+ \rightarrow product\ oxidation \tag{8}$$

$$P4R + e_{cb}^- \rightarrow product reduction \tag{9}$$

The $TiO_2G1HT$ oxide, present the best photocatalytic performance ($k_{app} = 5.9 \times 10^3$ $min^{-1}$; $R = 0.9824$), because the availability of reactive species becomes proportionally higher as the concentration of the dye decreases, since the concentration of these species is practically constant during the photocatalytic process (França et al., 2016). Therefore, P4R undergo fragmentation at the beginning of the reaction (Fig. 6 - Inset), which should favor a faster mineralization.

Analyzing the spectrum presented in the Inset of Fig. 6, it can be seen that at the end of the photocatalytic process, the band centered at 500 nm, referred to an electronic transition with a major component $\pi \rightarrow \pi^\star$ (Oliveira et al., 2012), involving the naphthalenic structures and the azo group, associated with the coloring of the dye, decreases significantly. The formed products should not present new or significant absorption bands in the analyzed region, suggesting that the degradation not only induces a quick discoloration of the dye (Table 2), as they should also cause a significant fragmentation of the dye structure, whose fragments should not absorb significantly in the monitored region of the electromagnetic spectrum.

The coexistence of anatase and brookite in the $TiO_2$ synthesized with different glycols and treated by a hydrothermal via at low temperature, minimized the recombination rate of the $e^-/h^+$ pairs, thus allowing the holes to be available for oxidation reactions. In addition, the correlation of physical and chemical factors, such as high surface areas and porosity, high photon absorption capacity in the UV-visible region and crystallinity considerably improved the photocatalytic activity of these oxides.

## CONCLUSIONS

In this study, we present the preparation of $TiO_2$ mesoporous nanoparticles using the sol–gel method with different glycols as structural molds. The use of ethylene glycol associated to further hydrothermal heat treatment proved to be the most effective way to obtain nanoparticles with improved photocatalytic activity. The results showed that materials submitted to hydrothermal heat treatment presented smaller particles and greater porosity, with formation of approximately spherical nanoparticles and with sizes up to 5 nm and formation of a binary mixture of anatase and brookite phases. The use of different glycols influenced the size of the particles, promoting the formation of smaller particles. The existence of a junction between different phases of the same semiconductor, accompanied by a decrease in the size of the particles, favored the charge transfer processes and contributed to the delay of the recombination processes, significantly improving the photocatalytic activity, verified by the degradation of the azo-dye Ponceau 4R under UV-Vis light irradiation. This type of photocatalyst that can harness both UV and visible light is a promising candidate for applications in photochemistry, sensors and solar cells, which has motivated us to develop oxides and nanocomposites based on $TiO_2$ with a wide spectrum of applications.

## ACKNOWLEDGEMENTS

The authors are thankful to Centro de Tecnologias Estratégicas do Nordeste (CETENE) and Laboratório Multiusuário de Microscopia de Alta Resolução (LABMIC) for the analysis of specific surface area and HRTEM images.

### Funding

This work was supported by the Brazilian research agencies Coordenação de Aperfeiçoamento de Pessoal de Nível Superior (CAPES) (PhD scholarships), Fundação de Amparo à Pesquisa do Estado de Minas Gerais (FAPEMIG CEX-APQ-00583-13 and CEX-APQ-03017-16), Conselho Nacional de Desenvolvimento Científico e Tecnológico (CNPq 307443/2015-9). The funders had no role in study design, data collection and analysis, decision to publish, or preparation of the manuscript.

### Grant Disclosures

The following grant information was disclosed by the authors:

Brazilian research agencies Coordenação de Aperfeiçoamento de Pessoal de Nível Superior (CAPES).

Fundação de Amparo à Pesquisa do Estado de Minas Gerais: FAPEMIG CEX-APQ-00583-13, CEX-APQ-03017-16.

Conselho Nacional de Desenvolvimento Científico e Tecnológico: CNPq 307443/2015-9.

## Competing Interests

The authors declare there are no competing interests.

## Author Contributions

- Lidiaine Maria Santos and Antonio Eduardo da Hora Machado conceived and designed the experiments, performed the experiments, analyzed the data, performed the computation work, prepared figures and/or tables, authored or reviewed drafts of the paper, and approved the final draft.

## Data Availability

Raw data are available in the Supplementary Files.

## Supplemental Information

Supplemental information for this article can be found online at http://dx.doi.org/10.7717/peerj-matsci.13#supplemental-information.

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
