# Peer review of "Synthesis of nano-TiO$_2$ assisted by glycols and submitted to hydrothermal or conventional heat treatment with promising photocatalytic activity"

_PeerJ Materials Science, doi:10.7717/peerj-matsci.13_

## Round 0.1 · original submission · Minor Revisions

The article seems to be suitable and merits publication (after incorporating the suggested changes).

Reviewer 1 ·

Basic reporting

No comment

Experimental design

No comment

Validity of the findings

No comment

Additional comments

The present manuscript titled "Synthesis of nano-TiO2 assisted by glycols and submitted to hydrothermal or conventional heat treatment with promising photocatalytic activity" is an interesting and compelling study. The results and discussion presented are interesting and can help other readers/researchers working in this area. I recommend the author(s) to address the following minor issues.

1 The caption in the inset of figure 4 mentioned the Optical "reflectance" spectra of the samples while in the figure it is indicated as absorbance spectra.
2. In figure 6, the linear fit in the case of TiO2 G1HT (A) doesn't seem right. Please correct it.
3. There are several typos in the manuscript. I recommend proofread the manuscript again.

Reviewer 2 ·

Basic reporting

The paper is well organized and clear. Literature were appropriately cited. The article is divided into different section and are appropriate

Experimental design

The experimental designs are appropriate and in line with the overall scope of the paper

Validity of the findings

The authors demonstrated the preparation of TiO2 mesoporous nanoparticles using the modified sol-gel method with different glycols as structure molds. The authors found the use of ethylene glycol associated with hydrothermal heat treatment proved to be the most effective way to obtain nanoparticles with improved photocatalytic properties. The use of different glycols influenced the size of the particles, promoting the formation of smaller particles. All the conclusions are well stated with the proper experimental studies. All data have been provided.

Additional comments

Overall quite good

Reviewer 3 ·

Basic reporting

The language for the most part is easy to understand. However, there are a few cases (listed in the comments to the author section) where the sentences are ambiguous.

Experimental design

Authors have attempted to compare the different heat treatment techniques in obtaining a better photocatalyst using a “modified” sol-gel method.
The research question is clearly stated. The authors have clearly stated the problem they are tackling and relate the experimental findings to the problem.
Also, the authors have performed several experimental techniques in the experimentation section, which are detailed in the discussion section.

Validity of the findings

The authors have done a remarkable job of performing several analyses to obtain structural information.
Many reference materials (including the analytical methods) are rightly cited by the authors.
Conclusions are clearly stated as well.

Additional comments

Authors have attempted to compare the different heat treatment techniques in obtaining a better photocatalyst using a “modified” sol-gel method.
As authors stated, Titanium dioxide has been attracting interest in various multidisciplinary applications.
In the introduction section, there are a few places there it is
The authors tried to explain recent applications of the TiO2 and the use of the language was very ambiguous. Some sentences and paragraphs do not make any sense even after going back-and-forth several times.
For example, lines 108 - 109
“….semiconductor oxides can to allow the displacement of electrons…”
The above sentence is not only hard to comprehend but also confuses the reader in grasping the authors’ views. There are several of these examples all over the manuscript.
Lines 111-112
“…Similar behaviors have the mixtures of crystalline phases of TiO2, such as anatase/brookite (A/B) or anatase/rutile (A/R)…”
The authors may want to restructure the above sentence.
Also, line 114-116
“In the present study, photocatalysts based on TiO2 were synthesized by the modified sol-gel method, evaluating the influence of the use of different structural molds (ethylene glycol, diethylene glycol or polyethylene glycol 300 ), as well as the effect of thermal treatments by conventional or hydrothermal route on the structural, optical and morphological properties. ”
It would be easier to break the above sentence and explain it effectively in two separate sentences.
Line 119-121
“…The results presented provide new ideas about obtaining mesoporous...”
please rewrite the sentence
Language should be improved to make sure that the international audience can clearly understand your manuscript.
Experimental methods:
Does using different structural molds will qualify to be the modified sol-gel?
In the results and discussion section:
In line 180:
The authors state that the formed agglomerates are spherical nanoparticles, which is conflicting with the TEM images.
In Fig. 1, D has a different scale compared to A-C, authors may want to replace it with an image with a similar scale.
Including the TEM and HRTEM images for TiO2G2M and TiO2G3M might help the reader to comprehend the Fig.1 better.
Authors may want to state the reference for their estimated spacing between crystalline planes an the .
Can the authors comment on why the Fig.2. (plot C had greater brookite phase than plot f)?

---

## Round 0.2 · accepted · Accept

The reviewers are satisfied with the improvements made through the revision and is ready to accept the manuscript for publication.

Reviewer 1 ·

Basic reporting

N/A

Experimental design

N/A

Validity of the findings

N/A

Additional comments

The author(s) have addressed the concerns raised previously. The revised manuscript looks good overall.